# Effect of Supplemental UV-A Intensity on Growth and Quality of Kale under Red and Blue Light

**DOI:** 10.3390/ijms23126819

**Published:** 2022-06-19

**Authors:** Haozhao Jiang, Yamin Li, Rui He, Jiehui Tan, Kaizhe Liu, Yongkang Chen, Houcheng Liu

**Affiliations:** College of Horticulture, South China Agricultural University, Guangzhou 510642, China; jhzh111@stu.scau.edu.cn (H.J.); yaminli@stu.scau.edu.cn (Y.L.); ruihe@stu.scau.edu.cn (R.H.); jiehuitan@stu.scau.edu.cn (J.T.); 1836945107@stu.scau.edu.cn (K.L.); cyk@stu.scau.edu.cn (Y.C.)

**Keywords:** kale, UV-A, antioxidant, growth, glucosinolate biosynthesis

## Abstract

Different intensities of UV-A (6, 12, 18 μmol·m^−2^s^−1^) were applied in a plant factory to evaluate the combined influences of supplemental UV-A and red and blue light (Red:Blue = 1:1 at PPFD of 250 μmol·m^−2^ s^−1^) on the biomass, antioxidant activity and phytochemical accumulation of kale. Supplemental UV-A treatments (T1: 6 μmol·m^−2^ s^−1^, T2: 12 μmol·m^−2^ s^−1^ and T3: 18 μmol·m^−2^ s^−1^) resulted in higher moisture content, higher pigment content, and greater leaf area of kale while T2 reached its highest point. T2 treatment positively enhanced the antioxidant capacity, increased the contents of soluble protein, soluble sugar and reduced the nitrate content. T1 treatment markedly increased the content of aliphatic glucosinolate (GSL), whereas T2 treatment highly increased the contents of indolic GSL and total GSL. Genes related to GSL biosynthesis were down-regulated in CK and T3 treatments, while a majority of them were greatly up-regulated by T1 and T2. Hence, supplemental 12 μmol·m^−2^ s^−1^ UV-A might be a promising strategy to enhance the growth and quality of kale in a plant factory.

## 1. Introduction

Considered as a healthy vegetable, containing abundant functional compounds, kale (*Brassica oleracea* var. *sabellica*) attracts more and more attention around the world. As a good source of minerals, vitamins, dietary fiber [1], vitamin C and particularly glucosinolates (GSLs) [2], kale is beneficial its anti-cancer properties.

In plant factories, light is the main factor that influences the nutritional quality of vegetables. The content and type of phenolic components in kale plants were strongly regulated by light conditions. Among the light sources used, RGB LED induced kale sprouts with the highest content of the main examined compounds (such as chlorophylls, β-carotene, lutein, neoxanthin and violaxanthin), to be obtained [3]. Higher contents of chlorophyll, protein, soluble sugar and antioxidant capacity were found in purple kale under low intensity light treatments (white or red) [4]. Among different light quality, UV-A brings remarkable effects on photosynthesis and the biomass of plants [5]. Compared with CK treatment (230 μmol·m^−2^ s^−1^ RB + 7 μmol·m^−2^ s^−1^ FR), supplemental low intensity UV-A (10 μmol·m^−2^ s^−1^) exerted remarkable effects on the contents of soluble sugar, soluble protein, carotenoid, total phenolic, vitamin C and total flavonoid, while higher intensity UV-A (40 μmol·m^−2^ s^−1^) presented a significant reduction of pigment content on lettuce at harvest [6]. In comparison with CK, broccoli sprouts yielded greater phenolic content under 3.16 W·m^−2^ UV-A (2 h) [7].

*Brassicaceae* plants are renowned for their prolific glucosinolates (GSLs) which is bound for their pungent flavor. By injuring the plant tissue, glucosinolates could be hydrolyzed by myrosinase enzyme to important degradation products (i.e.: isothiocyanates, nitriles and thiocyanates) [8]. Glucosinolates are sulfur- and nitrogen-rich secondary metabolites which could be distinguished into aliphatic GSL (AGS), indole GSL (IGS), and aromatic GSL(RGS) according to their amino acid precursors [9,10]. The desulfo glucosinolate fraction from *Lepidium perfoliatum* seed significantly reduced Reactive Oxygen Species (ROS) and Malondialdehyde (MDA) levels in liver mitochondria, which showed a significant hepatoprotective effect [11]. Blue light and UV-A at different intensities or wavelengths enhanced the GSL accumulation in red-leaf and green-leaf pak choi [12]. UV-B (1.5 kJ·m^−2^) retarded senescence and enhanced indolic GSL content of broccoli florets, which was consistent with the results of over-expression of *CYP79B3* [13]. The exposure to 1.2 kJ·m^−2^ UV-C extended shelf-life, significantly enhanced indole-type glucosinolates and hydrocinnnamates, lowered ascorbic acid contents, lowered oxygen radical absorbance capacity (ORAC) value, and promoted the higher expression of *CYP79A2*, *CYP79B3* and *CYP79F1* in broccoli florets [14]. The accumulation of secondary metabolites was improved and GSL-related genes’ expression (*MYB28*, *BCAT4*, *CYP79F1*, *CYP83A1* and *AOP3*) were mainly up-regulated in broccoli sprouts under R5B5 (red:blue = 5:5) treatment [15]. About seven genes (*MYB28*, *MYB51*, *CYP79F1*, *CYP83A1*, *Sur1*, *UGT74B1* and *UGT74B1*) related to GSLs biosynthetic in Chinese kale were down-regulated by supplemental FR [16].

Several studies have demonstrated that red and blue light have significant effects on the growth and quality of vegetables. UV-A have proved to be beneficial for secondary biosynthesis of vegetables, but related studies remain limited. Is there any interactive effects of UV-A and red and blue illumination on the growth and phytochemical accumulation in kale? And what would kale be under a different UV-A condition? In this study, different intensities of UV-A (T1: 6 μmol·m^−2^ s^−1^, T2: 12 μmol·m^−2^ s^−1^ and T3: 18 μmol·m^−2^ s^−1^) were applied in an artificial lighting plant factory to evaluate the influences of supplemental UV-A on biomass, antioxidant activity and phytochemical accumulation of kale under red and blue light.

## 2. Results

### 2.1. Growth and Biomass

The growth and biomass of kale were signally affected by different supplemental UV-A intensities (T1, T2 and T3) (Table 1 and Figure 1). Supplemental UV-A light treatments (T1, T2 and T3) exerted higher fresh weight, dry weight and moisture contents in shoot. The T2 treatment yielded the highest fresh weight and dry weight of roots, moisture content and total leaf area of plants, being 12.9%, 20.0%, 55.6% and 25.8% higher than CK, respectively. Higher specific leaf weight was observed in T2(~17.91%) and T3 (~21.34%) treatments, which indicated thicker leaves than CK.

In total, supplemental UV-A treatments promoted the growth and biomass of kale, while T2 treatment markedly exhibited the maximum increase.

### 2.2. Pigment Content

Regardless of supplemental UV-A treatments, the contents of chlorophyll a and chlorophyll b obviously increased while carotenoid content and chlorophyll a/b remained unchanged (Figure 2). The kale exposed to supplemental UV-A treatments (T1, T2 and T3) indicated higher contents of chlorophyll a than CK, with increases of 18.9%, 13.3% and 17.8%, respectively. Higher contents of chlorophyll b under T1, T2 and T3 (34.5%, 20.7% and 37.9%) treatments than CK were observed. Meanwhile, UV-A supplementation (T1, T2 and T3) promoted higher chlorophyll a + b contents, with 22.31%, 14.18% and 22.21%, respectively. Hence, supplemental UV-A treatments contributed to improve the pigment content of kale. 

### 2.3. The Contents of Soluble Protein, Soluble Sugar and Nitrate

The contents of soluble protein, soluble sugar and nitrate differed among light treatments (Figure 3). T2 treatment significantly increased soluble protein content while lower (T1) or higher (T3) intensity UV-A markedly decreased. Higher soluble sugar contents were obviously observed in T1 and T2 treatments, with increases of 29.4% and 27.9%, respectively. However, supplemental UV-A treatments facilitated a massive reduction of nitrate contents, about 36.5%, 43.3%, 15.9%, respectively, while the minimum nitrate content was recorded in T2 and the maximum was found in CK.

The exposure to supplemental UV-A treatments (T1 and T2) promoted higher contents of soluble sugar and soluble protein and lower nitrate content.

### 2.4. Antioxidant Capacity and Antioxidant Compounds 

The antioxidant capacity (DPPH and FRAP) and antioxidant compounds (total flavonoids, total phenolic, vitamin C and anthocyanin) contents under different light treatments were measured (Figure 4). There were insignificant differences in FRAP, in the contents of vitamin C, and in total phenolic among treatments. UV-A supplementation (T1 and T2) increased DPPH and total flavonoids content. However, a significant reduction of anthocyanin contents under UV-A supplementation treatments occurred.

### 2.5. Glucosinolate Content

Seven GSLs were extracted and detected in kale (Figure 5), with three aliphatic GSLs: sinigrin (SIN), glucobrassicanapin (GBN), gluconapin (GNA) and four indolic GSLs: 4-hydroxyglucobrassicin (4OH), 4-methoxyglucobrassicin (4ME), glucobrassicin (GBS), and neoglucobrassicin (NEO). The indolic GSL possessed 91.6% of the total GSL under CK, while its proportion was massively induced by T1 and T2, with increases of 16.2% and 41.1%, respectively. Overall, T1 markedly increased the content of aliphatic GSL, while T2 treatment promoted higher contents of indolic GSL and total GSL than CK. Supplemental UV-A treatments (T1, T2 and T3) enhanced SIN content, with increases of 30.4%, 48.7% and 11.5%, respectively. Compared with other treatments, T2 also exerted remarkable effects on the contents of 4ME and NEO, increasing by 31.29% and 79.02%, respectively. However, the total GSL content was slightly lower, by ~8.5%, under T3.

Hence, it is feasible that T2 contributed to the GSL biosynthesis.

### 2.6. The Gene Expression of Key Enzymes and Transcription Factors Related to Glucosinolates Biosynthetic Pathway

The expressions of genes related to the glucosinolate biosynthetic pathway were affected by supplemental UV-A light treatments (Figure 6). In transcription factor (TF) and side-chain elongation, T1 upregulated *MYB28*, *LeuC1*, *LeuD1* and *BCAT-4*, which were mostly related to the chain elongation of Met. T2 up-regulated *MYB51* and *LeuC1* which were part of the indolic GSL pathway. In addition, T3 upregulated *TFL2*.

In the core structure formation of GSL, nine genes (*CYP79F1*, *CYP83A1*, *SUR1*, *UGT74B1*, *UGT74C1*, *STa5*, *GSTF11*, *GSTF9* and *GGP1*) were upregulated by T1, while five genes (*SUR1*, *STa5*, *GSTU20*, *GSTF10* and *GGP1*) were significantly upregulated by T2. Most of the genes related to the GSL’s core structure formation were markedly down-regulated under CK.

In the side-chain modification of GSL, in which the individual GSL was produced, four genes (*CYP81F1*, *GSH1*, *MVP1* and *GSL-OH*) revealed signally higher expression in T2. Furthermore, CK and T1 slightly up-regulated the genes (*ESM1* and *ESP*) related to GSL degradation.

Hence, T1 and T2 facilitated higher expressions of the genes related to GSL biosynthesis, which was consistent with glucosinolate content in kale under different UV-A light treatments (Figure 5).

### 2.7. Heatmap Assay

A heatmap provided an integrated view of different intensity supplemental UV-A light treatments on the quality of kale (Figure 7).

The cluster exhibited different qualities of kale in different intensities of UV-A light treatments and CK at harvest. The kale in CK revealed a higher value of DPPH and higher contents of 4OH, GBN, soluble protein, nitrate, carotenoid and anthocyanin. However, the kale exposure to supplemental UV-A treatments demonstrated higher fresh and dry weight of shoot and root, higher biomass and specific leaf weight, higher contents of chlorophyll a and chlorophyll b, but lower nitrate content. T1 promoted the contents of SIN, GBN, GBS, vitamin C, soluble sugar as well as the value of DPPH and FRAP. T2 massively enhanced the contents of SIN, 4ME, NEO, indolic GSL, total GSL, total flavonoids and soluble protein, while T3 slightly increased the contents of 4OH, GBN, and total phenolic. These results indicated that supplemental UV-A light treatments contributed to enhance the growth and quality of kale.

### 2.8. Multivariate Principal Component Analysis

To compare the correlation of all quality characteristics of kale in supplemental UV-A treatments and CK, principal component analysis (PCA) was performed (Figure 8). The first six principal components, PC1–PC6 (eigen values > 1), account for 85% and 91% of the cumulative variance.

The first two factors (PC1 vs. PC2) were presented and revealed 58.2% of the total variance of kale in supplemental UV-A treatments and CK at harvest (Figure 8). CK was clearly separated from T1, T2 and T3, which validated that the results obtained under CK were significantly different from the supplemental UV-A treatments. Furthermore, T1 was close to T2, which meant these two treatments indicated familiar results, such as: GBS, SIN, DPPH, soluble sugar, total phenolic, fresh and dry weight of shoot and root, etc. PC1 was positively correlated to flavonoid, DPPH, carotenoid, NEO, soluble sugar, GBS, FRAP, root fresh weight, total leaf area, root dry weight and SIN, while it was negatively correlated to nitrate and GBN. PC2 showed a negative correlation to shoot dry weight, plant moisture content, shoot fresh weight, chlorophyll a, chlorophyll b, phenolic and specific leaf weight, and it positively correlated to vitamin C, anthocyanins, 4OH, GNA and soluble protein. The results indicated the relationship among nutrition, pigment, GSL and antioxidants by confirming the angle between two vectors (0° < positively correlated < 90°; uncorrelated, 90°; 90° < negatively correlated < 180°). Strong positive correlations were found between 4ME, DPPH, carotenoid, GBS, FRAP, root fresh weight, total leaf area, root dry weight and SIN of the kale, as their angles were less than 90°.

## 3. Discussion

### 3.1. Supplemental UV-A Affected the Biomass, Morphology and Pigment in Kale

To respond to changing light conditions, plants incur morphological and photosynthetic changes, and these responses depend on the light quality and intensity [17]. In this study, there were positive effects on the shoot height, SPAD, stem diameter, specific leaf weight and leaf area as a result of the UV-A treatments, especially the T2 treatment (Table 1). Moreover, there were positive correlations between biomass and total leaf area/specific leaf weight (Figure 8). Furthermore, supplemental UV-A treatments possessed higher moisture content than CK. Tomato seedlings demonstrated increases in the length of shoot and root, cotyledon leaf area, and fresh and dry matter content in response to UV-A 2H (0.45 J·m^−2^ s^−1^) [18]. The exposure to UV-B + UV-A of *Quercus robur* leaves indicated the highest leaf thickness, about 12.2% thicker than control [19]. UV-A could increase concentrations of chlorophyll and carotenoid, yielded greater leaf size, higher stem length, fresh and dry mass, and increased biomass in lettuce [20]. UV-A (18 µmol·m^−2^·s^−1^) increased leaf size and stem length and the fresh and dry mass of baby leaf lettuce [21]. Hence, suitable UV-A intensity (T2) massively increased the leaf area and specific leaf weight, which might induce photosynthesis and contribute to the higher biomass of kale.

In vegetables, light irradiation could trigger specific physiological processes resulting in pigment changes [22]. In this study, T1 and T3 sharply induced photosynthetic pigment synthesis which led to significantly higher contents of chlorophyll a + b (Figure 2), and, eventually, greater fresh and dry weight (Table 1). Low UV-A irradiation (10 µmol·m^−2^s^−1^) significantly increased chlorophyll contents in both lettuce cultivars (Yanzhi and Red butter) [23]. Pigment content was found to be massively increased in lettuce exposed to 0.45 J·m^−2^ s^−1^ UV-A [18]. However, a higher intensity of UV-A (40 µmol·m^−2^s^−1^) led to greatly lower pigment content in lettuce [6]. Thus, the effects of UV-A on pigment content of kale depend on the UV-A intensity. The biomass, morphology and pigment revealed a highly positive correlation (angle < 90°) (Figure 8), which meant that supplemental different intensity of UV-A promoted pigment biosynthesis, enlarged the leaf area (expanded photosynthetic area and improved photosynthetic efficiency) and eventually led to a higher fresh weight. 

### 3.2. Supplemental UV-A Affected the Kale Nutritive Quality

Soluble sugar and soluble protein are considered to be sensitive to light. Through unique enzymes (sucrose phosphatesynthase and sucrose-6-phosphate phosphatase), plants could transform starch and other primary carbon assimilation products into glucose and maltose during periods of light and eventually transported out of the plastid during the dark period via plastidic glucose and maltose transporters and entered the cytosolic sucrose biosynthetic pathway via the level of hexoses, and led to higher soluble sugar content [24]. Higher soluble sugar contents of kale were found under T1 and T2 treatments, while the highest soluble protein content was observed under T2 treatment (Figure 3). Lettuce leaves grown under high UV-A (40 µmol·m^−2^s^−1^) intensity had lower soluble sugar content compared to control, while low UV-A (10 µmol·m^−2^s^−1^) reached the highest contents of soluble sugar and soluble protein [6]. Supplemental UV-A (6 µmol·m^−2^s^−1^) increased 1 higher maltose in red leaf lettuce by a factor of 1.76 [25]. Supplemental UV-A treatment (40 µmol·m^−2^s^−1^) increased the contents of total soluble proteins, total phenolic compounds, and FRAP of broccoli microgreens more than CK (red:blue:green = 1:1:1, 30 µmol·m^−2^s^−1^) [26]. In red- and green-leaf pak-choi, the highest soluble protein contents were presented under supplemental 100 µmol·m^−2^s^−1^ UV-A (380 nm), which were 108.16% and 184.52% higher than CK under glasshouse, respectively. However, supplemental UV-A (380 nm and 400 nm) revealed a massive reduction of soluble sugar content in red- and green-leaf pak-choi [27].

Light quality is a key factor in regulating the nitrate content in plants. In this study, supplemental UV-A treatments exhibited a massive reduction of nitrate content, about 36.51%, 43.25% and 15.87%, respectively. The nitrite content exposed to supplemental UV-A (6 µmol·m^−2^s^−1^) decreased in red-leaf lettuce but massively increased in green-leaf lettuce [25]. However, supplemental UV-A (380 nm and 400 nm) demonstrated a sharp reduction in the nitrate contents of red- (~59.04% and ~31.65%) and green- leaf (54.00% and ~38.99%) pak-choi [27]. The soluble sugar contents were negatively correlated with the nitrate contents [28], which highly matched the results in this study (Figure 9). Suitable intensity UV-A (T2) promoted soluble sugar and soluble protein biosynthesis and intensified nitrate consumption by triggering photosynthesis.

Plants developed UV-absorbing compounds to avoid the damage to DNA or physiological processes caused by UV radiation, such as flavonoids and related phenolic compounds [29,30]. The induction of phenylalanine ammonia lyase (PAL), chalcone synthase (CHS) and other enzymes involved in the biosynthesis of flavonoids and other secondary compounds provided a valuable mechanism for coping with increased UV stress. These stimulations might eventually increase the levels and proportion of different polyphenols and flavonoids in the crop [31]. In this study, T1 and T2 treatments highly increased DPPH and flavonoids content, while T3 possessed the minimum amount, which indicated that higher intensity UV-A might weaken the antioxidant capacity, whereas, there was no significant difference of vitamin C content between treatments. Higher intensity of UV-A (T3) significantly decreased the anthocyanin content in kale (Figure 4). The contents of flavonoid, polyphenol, vitamin C, and total anthocyanin in two lettuce cultivars (Yanzhi and Red butter) were significantly higher under 10 µmol·m^−2^s^−1^ UV-A treatment [23]. The highest contents of total phenolic and total flavonoids were found in lettuce under low UV-A treatment (10 µmol·m^−2^s^−1^) after 15 days of storage compared to CK [6]. Extensive studies showed that the larger accumulation of flavonoids under UVB and UV-A suggested a protective role against UV damage; these compounds accumulated in leaves of higher plants to screen out harmful UV radiation [32]. Lower intensities of UV-A were beneficial for growth and anthocyanin accumulation and leaf coloration in baby leaf lettuce [21,33]. However, the anthocyanin content of Lollo Rosso lettuce was highly reduced under UV380, about eight times lower than those under UV280 [34].

### 3.3. Supplemental UV-A Affected Glucosinolate Biosynthesis in Kale

The consumption of *Brassica* (cruciferous) vegetables (broccoli, cabbage, kale etc.) has been reported to be effective against cancer, which might be due to a relatively higher content of GSLs [35]. Meanwhile, the biosynthesis of GSL is strongly related to supplemental light. In this study, UV-A could induce the accumulation of GSL. Supplemental UV-A treatments (T1, T2 and T3) massively increased the NEO contents. The accumulation of SIN, 4ME, NEO were highly induced by T2, whereas, the GBN content in kale grown under T2 presented the highest reduction. The indolic GSL possessed 91.6% of the total GSL under CK, while its proportion was positively induced by T1 and T2 (Figure 5). The broccoli exposed to white fluorescent tubes contained significantly higher contents of glucoraphanin, glucobrassicin, neoglucobrassicin, and 4-methoxyglucobrassicin than that in darkness after five days of storage [36]. Specific UV-B (0.9 kJ·m^−2^) mediated induction of GSLs, especially of 4-methylsulfinylbutyl GSL and 4-methoxy-indol-3-ylmethyl GSL in broccoli sprouts [37]. Low UV-A radiation (3.16 W·m^−2^) at harvest 2 h afterwards accumulated higher gallic acid hexoside I (14%), 4-O-caffeoylquinic acid (42%), gallic acid derivative (48%) and 1-sinapoyl-2,2-diferulolyl-gentiobiose (61%) [7]. Light exposure significantly increased the chlorophyll content, indicating that chloroplasts might exist at greater numbers and at higher vitality under light than under darkness. This might cause more aliphatic glucosinolates accumulated in baby mustard under light treatment [38]. Similarly, the positive correlation between pigment and GSL could be found in this study (Figure 8). Overall, these results revealed that suitable intensity supplemental UV-A treatments (T1 and T2) enhanced GSL biosynthesis, while higher intensity UV-A (T3) signally lowered the GSL content, and T2 might be the best suitable supplemental UV-A intensity.

GSLs and their degradation products play important roles in human body health, and the genes’ expression in GSL biosynthesis pathways are greatly regulated by light. In this study, in side-chain elongation and transcription factor, *MYB28*, *LeuC1*, *LeuD1* and *BCAT-4* were highly up-expressed under T1, while *MYB51* and *LeuC1* showed greater up-expression in T2. *MYB34*, *MYB51* and *MYB122* have been proved to be the magnificent transcription factors of indolic GSL biosynthesis [39].

With regard to core structure formation, the expressions of *CYP79F1*, *CYP83A1*, *SUR1*, *UGT74B1*, *UGT74C1*, *STa5*, *GSTF11*, *GSTU20*, *GSTF9* and *GGP1* were higher under low intensity UV-A (T1). Meanwhile, five genes (*SUR1*, *STa5*, *GSTU20*, *GSTF10* and *GGP1*) revealed higher expression in T2. Purple LED lights (60 µmol·m^−2^s^−1^, 420 nm) induced glucoraphanin (3.66-fold) and glucoerucin (10.56-fold) accumulation via up-regulating biosynthetic genes’ expression of aliphatic glucosinolates (*CYP79F1*, *FMO_GS-OX1_* and *AOP2*) in four-day-old broccoli sprouts [40]. The tryptophan N-hydroxylase (*CYP79B3*) were sixfold over-expressed to activate the GLS pathway by low UV-B doses (1.5 kJ·m^−2^), while threefold over-expressed dihomomethionine N-hydroxylase (*CYP79F1*) by high UV-B doses (7.2 kJ·m^−2^) [13]. UV-B treatments (0.3 and 0.9 kJ·m^−2^) triggered the transcription of *CYP71A*, *CYP71B*, *CYP79F1*, *CYP81F2*, *MAM3* and *BACT4* in broccoli sprouts [37].

In side-chain modification, the expression of *CYP81F1*, *GSH1*, *MVP1* and *GSL-OH* were significantly up-regulated in T2. *CYP81F2* catalyzes the hydroxylation of indol-3-ylmethyl GSL to 4-hydroxy-indol-3-ylmethyl GSL in *Arabidopsis thaliana* [41]. In addition, the GSL degradation genes *ESM1* and *ESP* might account for the reduction of GSL.

Under suitable intensity of UV-A (T1 and T2), the expression of genes related to GSL biosynthesis would be greatly up-regulated and eventually augment the GSLs contents (Figure 5).

## 4. Materials and Methods

### 4.1. Plant Materials and Growth Conditions

This experiment was carried out in the artificial lighting plant factory of the South China Agricultural University, Guangzhou, China. Seeds of kale (*Brassica oleracea* var. *sabellica* DC.) cv ‘Jingyu No.2’ (Beijing Jingyan Yinong Sci-Tech Development Center, Beijing, China) were sown into a moist sponge block and kept in the dark germination chamber for two days. After that, the germinated seeds in the sponge block were transplanted in the deep flow technique system with 1/2 strength Hoagland solution. Temperature was 21 ± 2 °C, CO_2_ concentration was 400–600 μmol·mol^−1^, relative humidity was 55–60%, EC ≈ 1.8 mS·cm^−1^, pH ≈ 6.4 and white LED lighting at 250 μmol·m^−2^s^−1^ PPFD from 6:00 a.m. to 6:00 p.m. After two weeks, the seedlings with two expended true leaves were transplanted into the plate (90 cm × 60 cm, 24 plants/plate).

### 4.2. Light Treatments and Sample Preparation

The LED panels (Chenghui Equipment Co., Ltd., Guangzhou, China; 150 cm × 30 cm) with red (660 ± 10 nm), blue (460 ± 10 nm) and supplemental UV-A (380 ± 10 nm) LEDs were used. After transplantation, the kale seedlings were planted under four treatments (6:00~18:00) with basal light (Red:Blue = 1:1 at PPFD of 250 μmol·m^−2^s^−1^): CK (basal light, non-UV-A treated), basal light + 6 μmol·m^−2^s^−1^ UV-A (T1), basal light + 12 μmol·m^−2^s^−1^ UV-A (T2), and basal light + 18 μmol·m^−2^s^−1^ UV-A (T3). All samples were collected at 21 days after UV-A supplementation (Figure 9).

The samples of kale leaf tissue for RNA extraction were collected free of any mechanical damage. Samples for phytochemical analysis were four replicates per treatment, and each replicate contained four plants. All of the samples were frozen in liquid nitrogen, then kept at −80 °C.

### 4.3. Biometric Measurements

The morphology indexes of kale were recorded at harvest. About seven kale plants in each treatment were randomly selected and weighed by an analytical balance. The samples were oven dried at 105 °C for 2 h and 70 °C for 72 h to determine the dry weight.
Moisture content (%) = (FW − DW)/FW × 100%

### 4.4. Pigment Content

Fresh leaf samples of kale (0.2 g) were soaked in a 6.0 mL acetone ethanol mixture (acetone:ethanol = 1:1, *v*:*v*) and incubated at 25 °C in the dark for 24 h. The extract solution absorbance was determined at 663 nm (A663), 645 nm (A645) and 440 nm (A440) with a UV-spectrophotometer (Shimadzu UV-16A, Shimadzu, Corporation, Kyoto, Japan). The pigment contents were calculated according to Lichtenthaler [42] as follows:Chl a content (mg/g FW) = (12.70 × A663 − 2.69 × A645) × 6 mL/(1000 × 0.2 g);
Chl b content (mg/g FW) = (22.90 × A645 − 4.86 × A663) × 6 mL/(1000 × 0.2 g);
Chl a + Chl b content (mg/g FW) = (8.02 × A663 + 20.20 × A645) × 6 mL/(1000 × 0.2 g);
carotenoid content (mg/g FW) = (4.70 × A440 − 2.17 × A663 − 5.45 × A645) × 6 mL/(1000 × 0.2 g)

### 4.5. Total Anthocyanins Measurement

The content of total anthocyanins (TA) was measured according to Rapisarda’s method [43]. Two fresh samples of kale (1.0 g) were homogenized with a potassium chloride buffer (50 mM KCl and 150 mM HCl, pH 1.0) and a sodium acetate buffer (400 mM CH_3_COONa and 240 mM HCl, pH 4.5), respectively. After five minutes of 4000× *g* centrifuging at 4 °C, the supernatants were determined at 510 nm with a UV-spectrophotometer.
TA (mg·g^−1^) = [(A1 − A2) × 484.8 × dilution factor]/24.825
where A1 and A2 are the absorbances of sample extracted from buffers of pH 1.0 and pH 4.5, respectively. The Figure 484.8 is the molecular weight of cyaniding-3-glucoside chloride, and the Figure 24.825 is the absorption coefficient at 510 nm. The dilution factor in this measurement is 1.

### 4.6. Phytochemical Measurements

#### 4.6.1. Soluble Protein Content Measurement

The soluble protein content was determined by Coomassie blue staining [44]. The 0.5 g fresh sample was ground with 8 mL distilled water and then centrifuged at 4000× *g* for 15 min at 4 °C. The 0.5 mL supernatant was diluted in 0.5 mL distilled water and well-mixed with 5 mL Coomassie brilliant blue G-250 solution. After five minutes, the solution was determined at 595 nm by a UV-spectrophotometer. Bovine serum albumin was used as a reference substance and the results were expressed in mg/g FW.

#### 4.6.2. Soluble Sugar Content Measurement

Soluble sugar content was determined with anthrone colorimetry [45]. The 1.0 g frozen fresh tissue was mixed with 10.0 mL deionized water, sealed with a plastic film, and boiled in 100 °C water bath for 30 min. Again, 10 mL deionized water was added, boiled in a 100 °C water bath for 30 min, and filtered by a funnel with double filter papers. The filtered solution was collected into a 25-mL measuring flask and cooled to 25 °C, then 25 ml of deionized water was added. Next, 0.2 mL filtered solution and 0.8 mL deionized water were mixed in a 20 mL test tube. 0.5 mL anthrone ethyl acetate reagent (Sinophaem, Beijing, China) and 5.0 mL concentrated sulfuric acid were added, mixed with vortex, and placed in a boiling water bath for 10 min. After cooling to ambient temperature, the solution was measured at 625 nm by UV-spectrophotometer, using deionized water as a blank. Sucrose (Guangzhou Chemical Reagent Factory, Guangzhou, China) was used as a reference substance and the results were expressed in mg/g FW.

#### 4.6.3. Vitamin C Content Measurement

Vitamin C content was determined according to molybdenum blue spectrophotometry [46]. The 0.5 g fresh frozen tissue was homogenized with 25 mL oxalic acid ethylene diamine tetraacetic acid solution (*w*/*v*) in a volumetric flask. The solution was filtered by a funnel with double filter papers. 10.0 mL supernatant was mixed with a 1.0 mL partial phosphoric acid-acetic acid solution (*w*/*v*) and 2.0 mL 5% sulfuric acid solution (*v*/*v*) and 4.0 mL 5% ammonium molybdate solution (*w*/*v*). The supernatants were mixed well and set still for 15 min, then measured at 705 nm by a UV-spectrophotometer, using oxalic acid-ethylene diamine tetraacetic acid as a blank. L-ascorbic acid (Guangzhou Chemical Reagent Factory, Guangzhou, China) was used as a reference substance and the results were expressed in mg/g FW.

#### 4.6.4. Nitrate Measurement

The nitrate content was determined by UV- spectrophotometer [47]. The 1.0 g fresh plant tissue was homogenized in 10 mL distilled water and heated in a boiling water bath for 30 min. The homogenate was filtered by volumetric flask. 0.1 mL sample solution was then mixed with 0.4 mL 5% salicylic and sulfuric acid and 9.5 mL 8% NaOH. The nitrate content was determined at 410 nm. Standards containing 0 to 120 μg NO_3_^−^-N in 1 mL aliquot were analyzed with each set of samples.

#### 4.6.5. DPPH Radical Inhibition Percentage Measurement

Fresh powder samples (0.5 g) were soaked with 8 mL ethanol for 30 min, then centrifuged 3000× *g* at 4 °C for 15 min. The DPPH radical inhibition percentage (DPPH) measurement was based on the method of Tadolini et al. [48]. After centrifugation, the supernatant was used to prepare three types of mixture (Ai: 2 mL Supernatant mixed with 2 mL 0.2 µM DPPH; Aj: 2 mL Supernatant mixed with 2 mL ethanol; and Ac: 0.2 µM DPPH mixed with 2 mL ethanol). These mixtures were determined at 517 nm by UV- spectrophotometer with a blank in similar manner of ethanol. The DPPH radical inhibition percentage was calculated as follows:DPPH (%) = [1 − (Ai − Aj)/Ac] × 100%

#### 4.6.6. Ferric Ion-Reducing Antioxidant Power Measurement

The FRAP assay was carried out according to Benzie and Strain [49]. The sample solution (0.4 mL), extracted following the same method as described for DPPH, was mixed with 3.6 mL of a solution containing 0.3 mol⋅L^−1^ acetate buffer, 10 mmol⋅L^−1^ 2,4,6-tripyridyl-S-triazine (TPTZ), and 20 mmol⋅L^−1^ FeCl_3_ at a 10:1:1 ratio (*v*/*v*/*v*) for 10 min at 37 °C. The FRAP of this mixture was determined at 593 nm using a UV-spectrophotometer. The FRAP calculated from a standard curve for Fe^II^ standard solution (FeSO_4_) was expressed as μmol g^−1^.

#### 4.6.7. Total Phenolic Content Measurement

The total phenolic content (TPC) was measured according to Rahman with slight modifications [50]. Fresh kale (0.5 g) was extracted with 8 mL alcohol for 30 min and then centrifuged 3000× *g* at 4 °C for 15 min. The supernatant (0.5 mL) was treated with 0.5 mL Folin–Ciocalteu’s phenol, and then 1.5 mL 26.7% Na_2_CO_3_ and 7 mL of distilled water was added. The mixture was incubated at room temperature for 2 h. The absorbance was determined by a UV–spectrophotometer at 760 nm. The TPC in each extract was determined and expressed as milligrams of gallic acid equivalents per gram of sample (mg GAE).

#### 4.6.8. Total Flavonoids Content Measurement

The total flavonoids content (TFC) was measured by the Al(NO_3_)_3_ colorimetric assay with slight modifications [51]. The sample solution (1 mL), extracted following the same method as described for total flavonoids, was added into 0.5 mL 30% methanol and 0.35 mL 5% NaNO_2_ solution. The mixtures were blended and kept for 5 min at 25 °C. After that, 10% AlCl_3_ (0.35 mL) was added, and 6 min later 5 mL 5% NaOH was added. The mixtures were kept at room temperature for 15 min. The sample absorbance was measured at 510 nm by a UV- spectrophotometer with a blank in a similar manner by replacing extract to alcohol. The TFC calculated from a standard curve for rutin was expressed as g L^−1^.

#### 4.6.9. Glucosinolates Contents Measurement

Glucosinolates were extracted and analyzed as previously described [16].

The samples were freeze-dried by vacuumed freeze dryer (ALPHA1-4, Osterode, Germany), and ground into powder. The powdered samples (200 mg DW) were extracted with 70% MeOH (*w*:*v*) and kept in a boiling water bath and ultrasonic bath for 15 min, respectively. After ultra-sonication, the extracts were centrifuged 3000× *g* at 4 °C for 10 min. The supernatants were collected for 5000 μL and mixed with 500 μL 500 mg L^− 1^ sulfatase solution, then stored in the Pasteur pipette columns (filled with 500 μL DEAE-Sephadex A-25). After 12 h, the desulfoglucosinolates solution could be eluted for HPLC. Sinigrin (Sigma-Aldrich, St. Louis, MO, USA) was used as an internal reference substance with their HPLC area and relative response factors (ISO 9167-1,1992) [47].

The glucosinolates were separated and identified by high-performance liquid chromatography (HPLC, Waters Alliance e2695). A 5 μm C18 column (Waters, 250 mm length, 4.6 mm diameter) was used for glucosinolate separation. Elution was performed with mobile phase A (water, 18.2 MΩ·cm resistance) and mobile phase B (acetonitrile). The optimum column temperature was 30 °C. At a flow rate of 1.0 mL/min, the gradient conditions were set as follows: solvent A volume at 100% for 0 to 32 min, 80% for 32 to 38 min, and solvent B volume at 100% for 38 to 40 min. The elution time was set for 42–50. The detector monitored glucosinolates at 229 nm. The individual glucosinolates were identified according to their HPLC retention times and our database. The results were expressed as μmol·g^−1^ DW.

### 4.7. RNA Extraction and qPCR

Total RNA was extracted using the RNAex Pro Reagent (Accurate Biotechnology Co., Ltd., Hunan, China). The RNA integrity was measured on a 1.2% denaturing agarose gel (*w*:*v*) using Agilent Bioanalyzer Model 2100 (Agilent Technologies, Santa Clara, CA, USA). The high purity RNA samples (1.8 < OD260/280 and OD260/230 < 2.2) were selected to construct cDNA, using Evo M-MLV RT for PCR Kit (Accurate Biotechnology Co., Ltd., Hunan, China).

The quantitative real time PCR (qRT-PCR) reactions were performed in a LightCycler 480 system (Roche, Basel, Sweitzer) with an Evo M-MLV RT-PCR kit (Accurate Biotechnology Co., Ltd., Hunan, China). The amplification was carried out as follows: the de-naturation at 95 °C for 30 s, followed by 40 cycles of amplification at 95 °C for 5 s, and annealing for 60 °C for 30 s. The melting curves were analyzed at the end of 40 cycles (95 °C for 5 s, followed by a constant increase from 60 to 95 °C). The relative gene-expression levels were normalized to the mean of *ACT* and calculated according to 2^−^^△△CT^ method [52]. qPCR reactions were prepared in three biological replicates. Primers used for this study could be found in Table 2.

### 4.8. Statistical Analysis

The measurements were carried out with three replications per treatment. The data was statistically analyzed with SPSS 23.0 (SPSS Inc., Chicago, IL, USA). The significance among the treatments were determined by analysis of variance (ANOVA) followed by Duncan’s test. The figures were made by OriginPro 9.0 (OriginLab Inc., Northampton, UK). TBtools [53] was used for the cluster heatmap.

## 5. Conclusions

The increasing awareness of kale as a phytochemical-rich vegetable has led to more studies aimed at improving phytochemical content. This study validated the biometric, morphological, photosynthetic, nutritive and antioxidant responses of kale to supplemental UV-A. Supplemental UV-A treatments contributed to yielded greater biomass production, higher pigment content, antioxidant capacity, soluble sugar, soluble protein and GSL content, and lower nitrate content. Meanwhile, red:blue = 1:1 at PPFD of 250 μmol·m^−2^s^−1^ supplemented by suitable intensity of UV-A (T1, 6 μmol·m^−2^s^−1^ and T2, 12 μmol·m^−2^s^−1^) could facilitate genes’ expression related to GSL biosynthesis. Thus, supplemental 12 μmol·m^−2^s^−1^ UV-A (T2) might be a valuable method to enhance the growth and quality of kale in a plant factory.

## Figures and Tables

**Figure 1 ijms-23-06819-f001:**
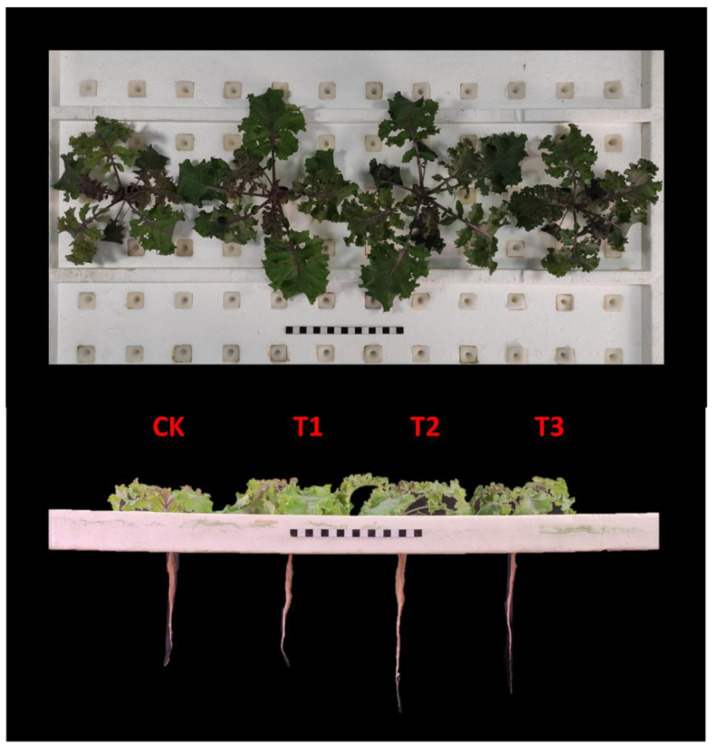
Kale samples at harvest under different UV-A treatments.

**Figure 2 ijms-23-06819-f002:**
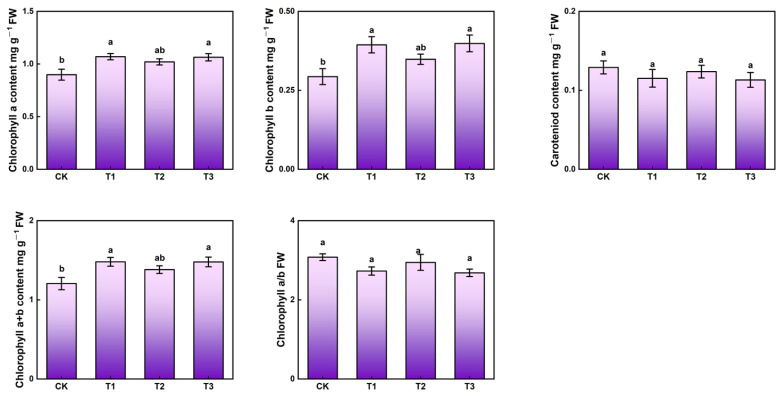
Pigment content in kale at harvest under different UV-A treatments. Values with different lowercase letters on the top of the columns indicate significant differences (*p* < 0.05), according to Duncan’s test. Vertical bars represent the standard margin of error.

**Figure 3 ijms-23-06819-f003:**
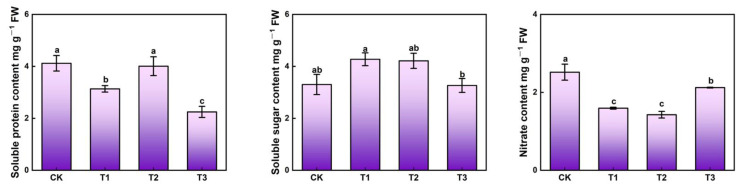
The contents of soluble protein, soluble sugar and nitrate in kale at harvest under different UV-A treatments. Values with different lowercase letters on the top of the columns indicate significant differences (*p* < 0.05), according to Duncan’s test. Vertical bars represent the standard margin of error.

**Figure 4 ijms-23-06819-f004:**
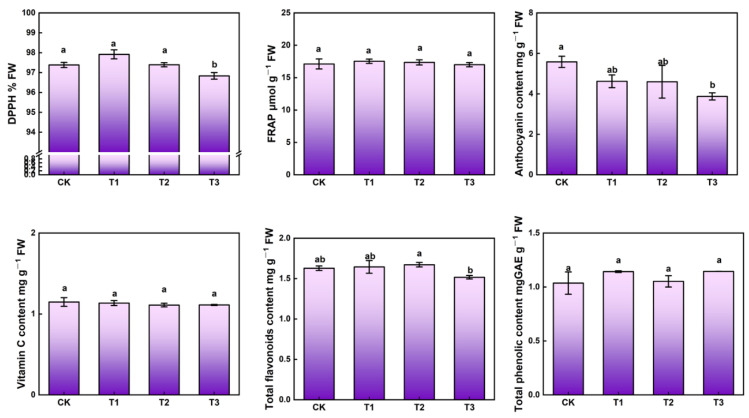
Antioxidant capacity and compounds in kale at harvest under different UV-A treatments. Values with different lowercase letters on the top of the columns indicate significant differences (*p* < 0.05), according to Duncan’s test. Vertical bars represent the standard margin of error.

**Figure 5 ijms-23-06819-f005:**
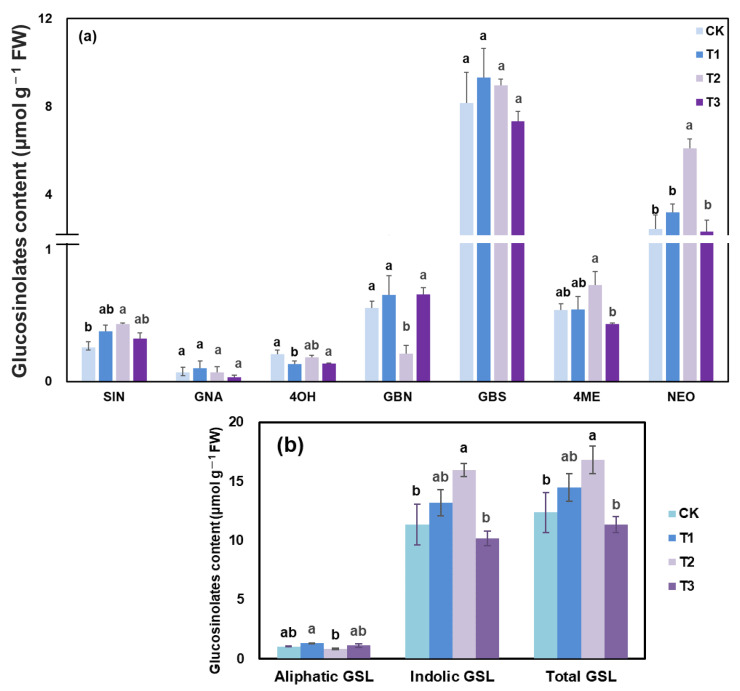
(**a**) seven individual glucosinolates content in kale at harvest under different UV-A treatments. (**b**) different sort of glucosinolate content in kale at harvest under different UV-A treatments. Values with different lowercase letters on the top of the columns indicate significant differences (*p* < 0.05), according to Duncan’s test. Vertical bars represent the standard margin of error. SIN = sinigrin, GNA = gluconapin, 4OH = 4-hydroxyglucobrassicin, GBN = glucobrassicanapin, GBS = glucobrassicin, 4ME = 4-methoxyglucobrassicin, NEO = neoglucobrassicin.

**Figure 6 ijms-23-06819-f006:**
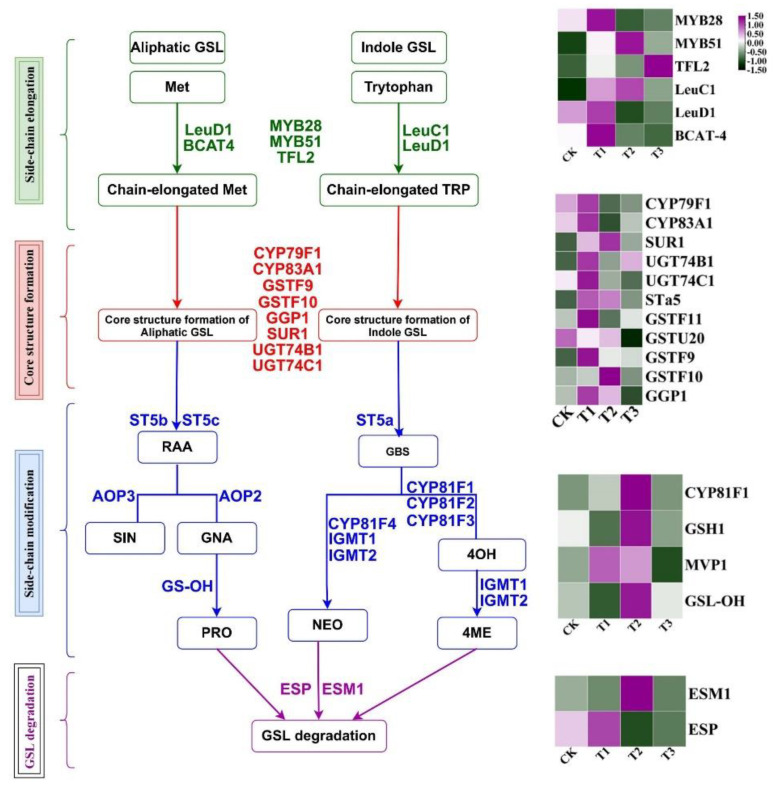
The genes’ expression related to glucosinolate biosynthesis and metabolism in kale under supplemental UV-A light treatments. Results are visualized using a false color scale with purple as an increase parameter, while green indicates a decreased parameter. The full name of GSL variants has been described in Figure 5.

**Figure 7 ijms-23-06819-f007:**
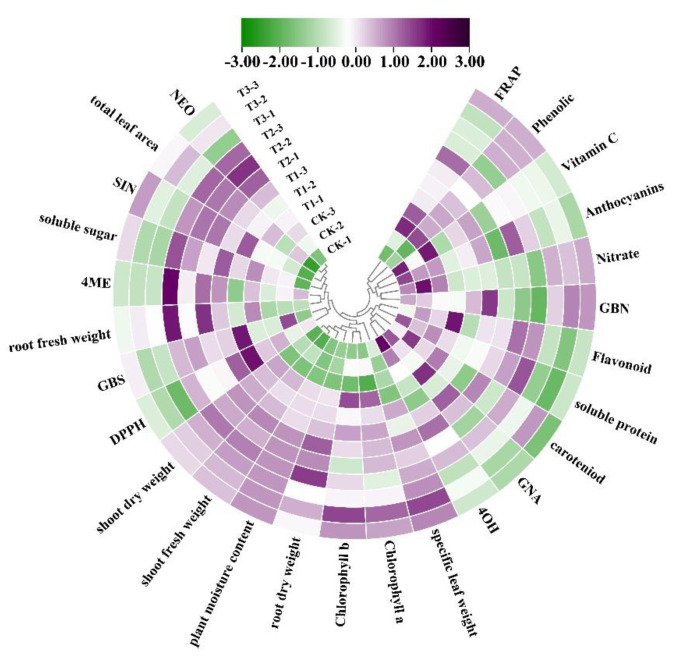
Cluster heatmap analysis summarizing the quality of kale in supplemental UV-A light treatments and CK. Results are visualized using a false color scale with purple as an increase parameter while green represents a decreased parameter. The full name of GSL variants has been described in Figure 5.

**Figure 8 ijms-23-06819-f008:**
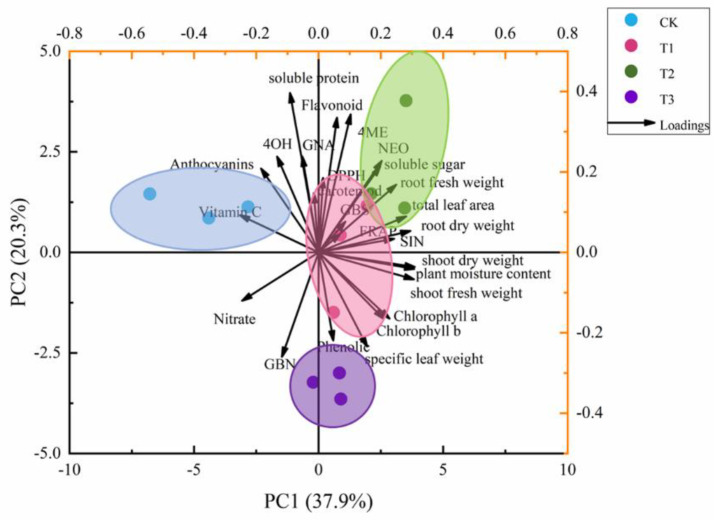
Multivariate principal component analysis showing supplemental UV-A enhanced quality of kale at harvest. The full name of GSL variants has been described in Figure 5.

**Figure 9 ijms-23-06819-f009:**
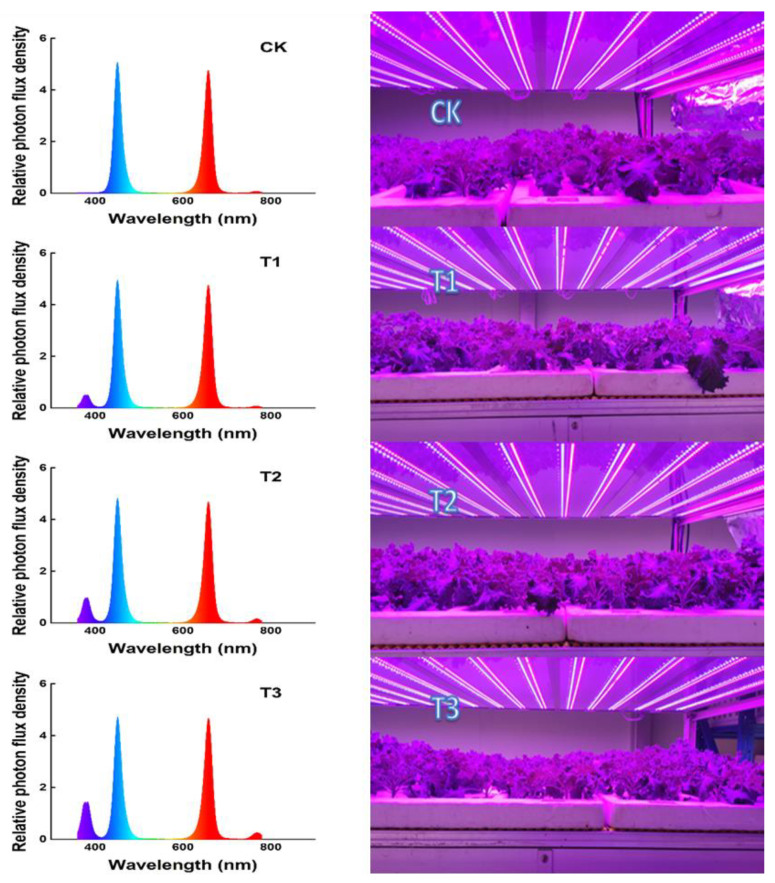
The spectrograms delivered by LEDs.

**Table 1 ijms-23-06819-t001:** The Growth of Kale at Harvest under Different UV-A Treatments.

Sample	Shoot Fresh Weigh g	Root Fresh Weight g	Shoot Dry Weight g	Root Dry Weight g	Plant Moisture Content%	Total Leaf Area cm^2^	Specific Leaf Weight mg/cm^2^
CK	14.94 ± 1.20 ^b^	3.93 ± 0.23 ^b^	1.28 ± 0.1 ^b^	0.65 ± 0.03 ^b^	89 ± 0.29 ^b^	359.92 ± 30.98 ^b^	2.624 ± 0.09 ^b^
T1	21.44 ± 0.97 ^a^	3.43 ± 0.22 ^ab^	1.78 ± 0.1 ^a^	0.71 ± 0.02 ^ab^	90 ± 0.24 ^a^	405.03 ± 9.76a ^b^	2.804 ± 0.18 ^ab^
T2	23.47 ± 1.42 ^a^	4.44 ± 0.56 ^a^	1.89 ± 0.1 ^a^	0.78 ± 0.03 ^a^	90 ± 0.23 ^a^	452.98 ± 24.80 ^a^	3.094 ± 0.15 ^a^
T3	21.20 ± 0.82 ^a^	3.23 ± 0.17 ^ab^	1.67 ± 0.0 ^a^	0.70 ± 0.02 ^b^	90 ± 0.14 ^a^	394.03 ± 11.78 ^ab^	3.184 ± 0.10 ^a^

Values with different lowercase letters (a and b) indicate significant differences (*p* < 0.05), according to Duncan’s test. “a” indicates more significant than “b”.

**Table 2 ijms-23-06819-t002:** Primers Used in QRT-PCR Analysis.

Gene	Forward primer (5′-3′)	Reverse primer (5′-3′)
ACT	TGGTTGGGATGGGACAGAAG	AATGCCGTGCTCAATAGGGT
MYB28	CGCTACTTCTGTTCTCGG	CGTTCTCCTCGTTGTGGT
MYB51	ATTATCGGAAGTGGTGGC	AGAATGTGGACGGAGACG
TFL2	GTGGGAATGTTGGTATGG	TAGCACCTATGAAACGACTG
LeuC1	GAGCGTTTGGTCAGTTTG	ACCTCATTGTTGGTGGAA
LeuD1	GCTGACAAAGCCACCATC	ACGACCCGCAACCAAAGT
BCAT-4	AGGCGTACAGGACAGAAG	TGGGATAAGGCATACAAAG
CYP79F1	TTAGACGAAGTGGTGGGA	GTGGCTACCTTTGGGAAT
CYP83A1	CTCCTTATCCCTCGTGCTT	CGTAGTCCGTGCCTTTGA
SUR1	TCGCCGATTATCTGAACC	GCGTCGTAGTGAGGGAAA
UGT74B1	CATCGACGCATACTCCGAATC	AACGGTGAGGTTGTTGGTGAAG
UGT74C1	GCAATGGCGATTAGACAA	CAGTGCGTCACGAAACAT
STa	TACCCGAGTCCATTGTCA	TAAGCCTTCCAGTAACCC
GSTF11	GTATGCGGACCAAGGAAC	TTGGGTGTAAAGACGATGTT
GSTU20	AACCCATTCTTCCCTTCT	CTGCTGCTTGTTCCTCAC
GSTF9	CACCACTTTCCACCCACC	CGTAGACATCGAGCACAGC
GSTF10	AGATCCCTGTGCTCGTTG	AAGCCATTGCTCTACTTGTC
GGP1	TCTTCGTCTTGGCTACAT	CTGGAACTCTGCCTTTGA
CYP81F1	CGTCCTTCCATACCTCCA	TCCTCCGTCGGTCTTCTA
ESP	CCTGAGGCTCGTACTTACC	AAACCACCCAAATCTTCC
MVP1	TTGTTCTACATCGGCTCTG	GGACCCGTAAATCACCTT
GSL-OH	TCCTGAGCCTGACCTAAC	AAGAACTTGAAGCCCACC
GSH1	ACAGAAGGAAAGCCAAAC	TACTGCTCAAACCCAAAA
ESM1	ATCGGCTTGTTATACTCCT	CAAAGACGGTGAACTGAA

## Data Availability

The datasets generated during the current study are available from the corresponding author on request.

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
