# Peer review of "Effect of Supplemental UV-A Intensity on Growth and Quality of Kale under Red and Blue Light"

_ijms, 2022, doi:10.3390/ijms23126819_

Round 1
Reviewer 1 Report
Manuscript titled: “ Effect of Supplemental UV-A Intensity on Growth and Quality of Kale under Red and Blue light” by Haozhao Jiang and coworkers, submitted to International Journal of Molecular Sciences, gives results on the interactive effect of UV-A on biomass, antioxidant activity and phytochemical accumulation of kale grown under red and blue light.
The main problem emphasized in this manuscript is an interactive and complementary effect of UV-A and red & blue illumination on growth and phytochemical accumulation in kale.
The Introduction gives an overview on treatment of different light quality, including UV-A, on several plant metabolites and processes such as chlorophylls, β-carotene, lutein, carotenoids, protein, soluble sugar, photosynthesis etc. Literature data on the influence of different light spectrum, are not consistent and depend more on the light intensity than on quality and also on the species examined. Based on the literature data it seems that supplementation of blue and red light treatment by UV-A can improve plant growth. Therefore there was a good reason to investigate further this subject. Kale (Brassica oleracea), an important from the agriculture point of view plant species and a healthy nutrient-rich plant is a good choice for such investigations.
My first remark is that the meaning of T1, T2 and T3 treatments; it is not explained early enough, neither in Summary, neither in Results, neither in first Table; but only in Material and methods at the end of the manuscript.
The authors showed that the supplemental UV-A promoted growth and biomass of kale, contributed to improve the pigment content, however the differences between UV-A and control were not large. The authors demonstrated that the supplemental UV-A treatments:T1 and T2 promoted higher content of soluble sugar and soluble protein and caused a lower nitrate content. The content of antioxidants changed depending on the compound and on the light spectrum; especially the UV-A supplementation caused a significant reduction of the anthocyanin content compared with control. The content of glucosinolate (GSL), secondary metabolites, typical for the Brassicaceae family, were also measured and it was shown that the T2 treatment contributed to the total amount of GSL biosynthesis.
Expression of different key enzymes and transcription factors related to the biosynthesis of glucosinolates were analyzed. According to the authors, T1 and T2 facilitated higher expression of genes related to the GSL biosynthesis, similar effect was observed in the glucosinolate content in kale under different UV-A light treatments.
I consider this part of investigations as the most interesting one. I want to point out, however, that the color scale used in charts related to this and the next part is confusing. In Fig 6. green indicates “a decreased parameter” and in the next Fig. 7 „ green as an increase parameters”. Also there is a mistake in the Fig. 6 caption. Please change in the legend of Fig 6 “The full name of GSL variants has been described in Figure 5”.
The heatmap (Fig. 7) summarizing the response of all measured parameters presents an overview of the supplemental UV-a treatment on the quality of kale. I consider this overview as the most valuable result of this work.
The correlation of all quality traits of kale in supplemental UV-A treatments and in control was found by the principal component analysis. This analysis indicated which traits are strongly correlated and which are negatively correlated. According to the authors most of measured parameters were similar in T1 and T2. I am sure that the overall analysis of the supplemental UV-A is promising and gives real indication for breeding enhancing the quality of kale growth. In my opinion such analysis brings a lot of new information and is not often done in biological research.
In conclusion, I highly appreciate the results of this work which validated the biometric, morphological, nutritive, photosynthetic, and antioxidant responses of kale to the supplemental UV-A treatment showing that it enhanced the quality of kale at harvest.
I recommend to publish the manuscript in the International Journal of Molecular Sciences after introducing minor corrections mentioned above.
Author Response
Thank you for your kind and professional suggestion! Please find the responses and revised manuscript in the attachment.

Reviewer 2 Report
This is an interesting paper, demonstrating the positive effect of supplemental UVA light on growth and phytochemical composition of kale, grown under artificial light. The topic is interesting, but the paper has some weak points, mainly in description of used methodology as follow:
1. The expression of DPPH results as % of inhibition was not appropriate to compare the observed changes in biomass. This type is more acceptable for comparing solutions of pure substances, whereas for this study, either IC50 or Trolox equivalents should be used.
2. The FRAP assay was described incorrectly. There is also no information about used standard.
3. The description of total phenolic assay was incomplete.
4. The description of total flavonoids assay was incomplete.
5. I cannot accept the described procedure for identification and quantification of glucosinolates. At least internal standards should be used.
6. There are no accession numbers or references to used primers for RT-qPCR.
Author Response
Thank you for your kind and professional suggestions! Please find the responses and revvised manuscript in the attachment.

Round 2
Reviewer 2 Report
The authors introduced some minor corrections but unfortunately, I cannot see any improve on scientific quality of the paper.
Author Response
Thank you for your kind and professional suggestion. Please find the response and revised manuscript in the attachment.
